# Promoting social distancing in a pandemic: Beyond good intentions

**Paolo Falco** [ORCID] [1]⊗*, **Sarah Zaccagni** [ORCID] [2]⊗

**1** Dept. of Economics, University of Copenhagen, Copenhagen, Denmark, **2** Dept. of Economics and Center for Economic Behaviour and Inequality, University of Copenhagen, Copenhagen, Denmark

⊗ These authors contributed equally to this work.
* paolo.falco@econ.ku.dk

**Data Availability Statement:** Pre-print and data are already available at this link: https://osf.io/a2nys/.

**Funding:** This study was funded by the following: Department of Economics of the University of Copenhagen, Department of Public Health of the

## Abstract

Do reminders to promote social distancing achieve the desired effects on behavior? Much of the existing literature analyses impacts on people's intentions to comply. We run a randomised controlled trial in Denmark to test different versions of a reminder to stay home at the beginning of the crisis. Using a two-stage design, we follow up with recipients and analyse their subsequent self-reported behaviour. We find that the reminder increases ex-ante intentions to comply when it emphasises the consequences of non-compliance for the subjects themselves and their families, while it has no effect when the emphasis is on other people or the country as a whole. We also find, however, that impacts on intentions do not translate into equivalent impacts on actions. Only people in poor health react to the reminder by staying home significantly more. Our results shed light on important gaps between people's intentions and their actions in responding to the recommendations of health authorities.

## Introduction

In the first months of 2020, a new type of coronavirus named SARS-CoV-2 began to spread like wildfire from China to the rest of the world. By November 2021, 246 million people had been infected and nearly 5 million people had died from the disease worldwide (Source: Worldometers website [link]).

In the absence of a cure or a vaccine, fighting a pandemic requires people to abide by certain norms of behaviour ([1]) and to follow the guidelines of authorities in a coordinated fashion ([2, 3]). Such recommendations span several domains, from personal hygiene to spending more time at home and avoiding contact with people who face the greatest risks ([4, 5]).

Social distancing—the practice of maintaining a physical distance between people and reducing the number of times people come into close contact with each other—is the most effective way of reducing contagion ([6–8]). It is also difficult to enforce. In its most extreme form, social distancing implies that people should remain in their homes and avoid contact with others, unless strictly necessary. Such strict forms of distancing have been applied in countries like Italy and France. Milder forms of social distancing have been encouraged across the globe.

University of Copenhagen, Center for Healthy
Ageing, Center for Economic Behaviour and
Inequality. The funders had no role in study design,
data collection and analysis, decision to publish, or
preparation of the manuscript.

**Competing interests:** The authors have declared
that no competing interests exist.

Since social distancing is disruptive for people's lives, authorities have been struggling to find ways of promoting it ([9]). Awareness campaigns have been numerous in many countries and reminders of different sorts have been used, ranging from social media campaigns to SMSs like those sent by the Danish Police to every mobile user (March 22) (Source: The Local DK (March 24, 2020) [link]) or the British government to every UK resident (March 24) (Source: UK Government website (March 24, 2020) [link]). In other contexts, such as smoking cessation, medical adherence ([10–13]), physical activity ([14]), seat belt usage ([15]), take-up of social benefits ([16]), electricity consumption ([17, 18]), and giving to charitable organizations ([19–21]), reminders have been shown to cause behavioural change ([22]).

Do messaging campaigns to promote social distancing achieve the desired objective? While few studies have shown that reminders affect people's ex-ante intentions ([23–26]), we know little about subsequent impacts. In light of the large literature documenting intention-to-action gaps and time inconsistency across a wide range of domains ([27–33]), discrepancies between intended behaviour and subsequent actions deserve investigation. To the authors' knowledge, this is the first paper to test the impacts of reminders to promote social distancing on both intended behaviour and subsequent actions.

We run a randomised controlled trial in Denmark to test different versions of a reminder to stay home at the beginning of the crisis. The reminders vary along two dimensions: pro-social motives and loss-gain frame. With regard to the first dimension, we hypothesise that emphasising the social proximity of those who would bear the consequences of non-compliance can increase the effectiveness of the message ([34]). With regard to the second dimension, we build on insights from prospect theory ([35]) and hypothesise that people may react differently to a framing that emphsises the potential losses from not complying as opposed to the potential gains from compliance ([36]). Using a two-stage design, we follow up with recipients and we can analyse subsequent impacts.

We find that the reminder increases ex-ante intentions to comply when it emphasises the consequences of non-compliance for the subjects themselves or their families, while it has no impact when the emphasis is on other people or the country as a whole. Our main finding, however, is that respondents largely do not follow through with their intentions, as the reminder has no significant impacts on subsequent behaviour. This is consistent with the existence of imporatnt intention-to-action gaps. While the behaviour we study is self-reported, this does not pose a threat to our conclusions, since potential mis-reporting would likely go in the direction of compliance being over-stated ([37]). Despite that, we find no impact of the reminder on behaviour, despite the effect on ex-ante intentions.

This study contributes to the literature on the effectiveness of reminders in promoting healthy behaviours ([10–15]). In particular, we relate directly to recent studies that have found impacts of reminders on people's intentions to comply with regulations to curb the spread of COVID-19 ([23–26]) and have investigated the determinants of the adoption of protective behaviour regarding COVID-19 ([38–41]). We complement the literature on the impacts of behavioural tools by showing that intentions to comply may fail to translate into equivalent behavioural change.

Our study is also of immediate policy relevance. By focusing on the decision to stay home, we test the effectiveness of a key recommendation provided by health authorities across the world. From the UK Prime Minister (Source: BBC (March 23, 2020) [link]), to the President of the United States (Source: The Washington Times (March 31, 2020) [link]), to the Queen of Denmark (Source: The Local DK (March 18, 2020) [link]), the advice to stay home as much as possible has been ubiquitous during the COVID-19 pandemic. Yet, convincing people to follow this recommendation is difficult, since it implies major changes to their routine and can

be perceived as a severe limitation of individual freedom. This study shed light on those challenges.

## Experimental design

We conduct a pre-registered randomised controlled trial with Danish residents aged 18–69 (Registry number AEARCTR-0005582 [link]). This study was reviewed and approved by the Research Ethics Committee of the Faculty of Social Sciences at the University of Copenhagen (application nr. 1504833) before it began. Upon receiving the invitation to participate by email, subjects were informed about the purpose of the study and they were asked for their consent (which they could give by clicking on a button on their screen). We expose different groups to different variations of a recommendation to "stay home as much as possible" and we test the impact of the treatment on both respondents' intentions to stay home the following day and on whether they report having stayed home. Our data, described below, closely track widely used mobility measures based on mobile-phone data, corroborating the reliability of the information on respondents' behaviour.

We test four alternative ways of framing the recommendation, extending previous research that investigates self-interested versus prosocial motives as drivers of compliance with health recommendations ([34, 42–46]). The first frame ("you") focuses on the potential consequences of the subject's behaviour for himself/herself. The second frame ("family") focuses on the consequences for his/her family. The third frame ("others") focuses on the consequences for other people in general. The fourth frame ("country") focuses on the broader consequences for the country as a whole by emphasising the risk of overloading the health system. This approach builds on existing studies showing that self-interest and emotional proximity to others (i.e., whether the person affected by one's decision is a stranger or a friend) is an are important determinants of people's choices ([47–49]).

For each of the four treatments we test two variations. The first one, in the loss domain, emphasises the risks of not complying with the recommendation (for the respondent, the family, others, and the country). The second, in the gain domain, emphasises the benefits of complying with the recommendation. This approach builds on prospect theory ([50]), which predicts that losses motivate behaviour more than equivalent gains. This is because in prospect theory decision making is based on value relative to a reference point and people typically weigh losses more than equal gains. The theory has found confirmation across a range of contexts. A classic example related to our experiment is that medical treatments described as having a "75% survival rate" are viewed more positively than those with a "25% mortality rate" ([51–53]). It follows that people are commonly more responsive to incentives framed as potential losses relative to an initial endowment than to equivalent gains relative to a reference point of 0. Recent work demonstrates this in the context of a smoking-cessation intervention where incentives framed as losses relative to an endowment had significantly stronger effects than incentives framed as equivalent gains relative to no initial endowment ([54]). A vast literature offers similar examples in different domains ([55–58]).

In addition to four framed messages, we send a generic reminder to stay home as much as possible without any framing. This is akin to the simple appeals made by health authorities and politicians in televised speeches and social media campaigns. Finally, a control group receives no reminder. Table 1 reports a summary of the treatments, including the text of the reminders. Each treatment appeared to the respondents as a text box with the message on a red background (see Appendix B in S1 Appendix).

**Table 1. The framing of reminders.**

| DOMAIN | FRAME | REMINDER |
|---|---|---|
| *You* | *Loss* | "If you go outside and become infected, you may get very serious respiratory problems. Stay home as much as possible." |
| | *Gain* | "If you stay home, you protect yourself from the risk of getting very serious respiratory problems. Stay home as much as possible." |
| *Family* | *Loss* | "Think of your loved ones. If you go outside and become infected, you may infect them, and they may get very serious respiratory problems. Stay home as much as possible." |
| | *Gain* | "Think of your loved ones. If you stay home, you protect them from the risk of getting very serious respiratory problems. Stay home as much as possible." |
| *Others* | *Loss* | "If you go outside and become infected, you may infect others, who may get very serious respiratory problems. Stay home as much as possible." |
| | *Gain* | "If you stay home, you protect others from the risk of getting very serious respiratory problems. Stay home as much as possible." |
| *Country* | *Loss* | "If you go outside and become infected, you may contribute to an overloading of the Danish health care system. Stay home as much as possible." |
| | *Gain* | "If you stay home, you reduce the risk of an overloading of the Danish health care system. Stay home as much as possible" |
| *Generic* | | "Stay home as much as possible." |

*Notes*: The table provides an overview of the reminders we sent. We tested four frames with a focus on "you", "family", "others", and "country", respectively. Approximately 6,000 subjects were assigned to each frame. They were equally split between two variants of the frame (loss versus gain). In addition, 3,000 subjects received a generic reminder with no framing and 3,000 subjects in the control group received no reminder.

## Theoretical framework

In this section, we outline a simple theoretical framework that guides our hypotheses. The model builds on Löfgren, A. and Nordblom, K. (2020) [59] and describes the choice of an individual between staying home ($x_1$) and going out ($x_2$). The individual chooses the option that yields higher utility, as described by the following equation:

$$V_i(x) = \theta E[U_i(x) + \alpha U_o(x)] + (1 - \theta)\mu_x, \quad x = x_1, x_2, \tag{1}$$

The individual's utility has two parts. $E[U_i(x) + \alpha U_o(x)]$ is expected utility defined as the sum of own utility $E[U_i(x)]$ and utility of others $E[U_o(x)]$ discounted by an altruism parameter $\alpha$. $\mu_x$, on the other hand, captures preference-irrelevant attributes. Total utility is a weighted sum of expected utility and utility from such attributes with the relative weight of the two determined by an *attentiveness* parameter $\theta \subset [0, 1]$ (also described as a *confidence* parameter). The higher the level of attentiveness/confidence with which an individual makes the decision, the higher the weight placed on expected utility as opposed to other preference-irrelevant attributes. In other words, more attentive individuals are the ones who think more carefully about their utility and the utility of others when making a decision.

Our hypothesis is that reminders act as a nudge that helps respondents to make more attentive choices (i.e., they increase $\theta$). We believe this is a better characterisation than assuming the reminders provided new information, given that basic knowledge about COVID-19 spread very quickly at the onset of the pandemic. Another way to describe our approach is to say that we consider reminders as *preference nudges*. Preference nudges have an impact on the expected utility in an inattentive choice situation, without altering the attentive choice ([59]). In standard utility theory, attentive choices are those in which individuals make an informed choice resulting in the preferred outcome. On the contrary, inattentive choices are based on heuristics

and therefore may lead to mistakes ([59]). We cannot exclude that some of our reminders may also increase $\alpha$, the weight people attach to other people's utility (i.e., their pro-social motives), but we do not aim to formally disentangle the two channels.

## Data collection and summary

The study covers the period between March 25 and April 7, 2020, a time when the first wave of the crisis was at its peak in Denmark and financial markets in turmoil, causing authorities to adopt stringent measures (including the closure of non-essential economic activities and of the borders), which however did not include an obligation to stay home as in other countries (e.g., Italy, France). Appendix E in S1 Appendix, provides a detailed timeline of the initial phase of the crisis in Denmark and shows that our study falls within the most critical period.

The different reminders were randomised within a representative sample of 29,756 Danish residents between the age of 18 and 69, who represent close to 1% of the population. A randomly selected control group received no reminder. Statistics Denmark (the national statistical office) carried out the randomisation and distributed the treatment via e-Boks—the official system of communication used by public authorities in Denmark, which is akin to a personal email account. By focusing on the general population of Danish residents, we avoid having to rely on selected panels of respondents on dedicated survey platforms, who may have been solicited frequently during the COVID-19 crisis and may be subject to fatigue ([60, 61]). Respondents received a message inviting them to participate in a survey to investigate people's habits at the time of the COVID-19 crisis. Upon receiving the invitation to participate, subjects were informed about the purpose of the study and they were asked for their consent (which they could give by clicking on a button on their screen). Those who agreed to participate landed on a dedicated webpage where they were first shown the reminder (if they were in one of the treated groups) and then answered eight questions (control subjects only saw the questions).

Out of the 29,756 subjects contacted by Statistics Denmark, a total of 12,573 (42,2%) completed the first survey. Out of those, 6,681 (22.3%) completed the second survey. The first questionnaire was sent to respondents on March 25 and the last responses to the second questionnaire were received on April 7. We drop 2 respondents whose answers to the questions on time spent home exceed 24 hours and 7 observations in which the follow-up refers to the day before the treatment occurred (which was the result of a technical problem). Furthermore, since some of the participants responded with some delay, it is possible that the answers to the follow-up questionnaire do not refer to the same day as the answers to the first questionnaire. To prevent major inconsistencies, we drop from the analysis respondents whose answers to the first questionnaire referred to a weekday, while their answers to the follow-up questionnaire referred to a day of the weekend, and vice-versa. This makes our results more precise, but does not change them qualitatively (as discussed below). This leaves us with a balanced panel of 5,310 respondents, which we use for the analysis. In Appendix F in S1 Appendix, we test whether the probability that a respondent drops out of the sample between the first and the follow-up survey correlates with assignment to treatment. Table 4 in S1 Appendix shows that attrition is balanced across treatment groups and does not depend on the kind of reminder respondents receive. As a result, covariates are balanced across treatments (see Table 4 and Appendix A in S1 Appendix).

While the first questionnaire asked about intentions to stay home *the day after*, the second questionnaire asked about whether the respondent went out *the day before* (both questionnaires are available in Appendix C in S1 Appendix). Specifically, we asked respondents how long they were planning to spend (first questionnaire) or did spend (second questionnaire)

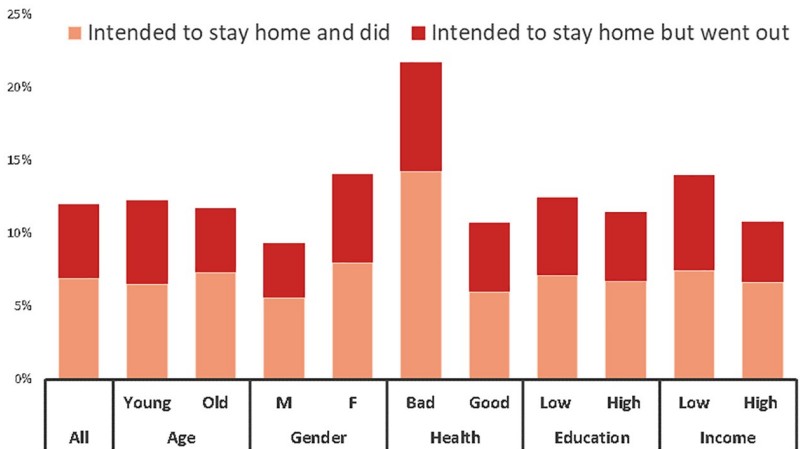

**Fig 1. The gap between intentions and actions.** *Notes*: 42% of the respondents who intend to stay home do not follow their intentions and go out instead. Intentions to stay home are higher among women and lower-income households, while they do not change significantly by age or education levels. People with relatively poor health conditions are the most likely to stay home and the least likely to deviate from such an intention. The sample for this figure is restricted to respondents whose answers to the first and the second questionnaires refer to the same day, since we are interested in documenting inconsistencies (N = 3,032). The subsequent analysis of treatment effects can rely on a larger sample since we do not need an exact match between the days. The different categories are defined as follows: young: < 50; low education: < post-secondary degrees; low income: household disposable income per capita < 250,000 DKK (approx. 36,000 USD).

outside their home, and we treated positive answers as instances of not staying home. We believe this is a preferable strategy to asking whether respondents were planning to go out (a "Yes/No" question), since it induced more careful reflection. The results are robust to treating respondents who only went out for a very short time (e.g., less than 5, 10, or 20 minutes) as having stayed home.

Since the information on actions (i.e., whether a person stayed home) is self-reported by the respondent in our follow-up survey, we use the data released by Apple Inc. on mobility trends during the COVID-19 pandemic to validate the reliability of this measure (Source: Mobility Trends Report by Apple Inc. [link]). Apple publishes daily reports on mobility trends based on mobile phone data, in countries and cities around the world. We extract the aggregate data for Denmark and we compare them with the distance that the subjects declared to have travelled the day before in our follow-up survey. To make the two series comparable, we calculate percentage changes in mobility compared to the first day for which we have information in our dataset. The results are reported in Appendix E and Fig 5 in S1 Appendix.

A summary of our data shows that less than 15% of respondents intend to stay home the next day during the most critical period of the first pandemic wave, and 42% of them do not follow the declared intentions and go out instead (Fig 1). Intentions to stay home are higher among women and lower-income households, while they do not change significantly by age or education levels. People with relatively poor health conditions are the most likely to stay home and the least likely to deviate from such an intention.

## Estimation strategy

To measure the effects of the reminders, we compare the probability of going out in each of our treatment groups with the probability of going out among respondents in the control group that received no treatment. More specifically, we estimate the following econometric

model:

$$Y_i = \beta_0 + \beta_1 T_{i,1} + \beta_2 T_{i,2} + \beta_3 T_{i,3} + \beta_4 T_{i,4} + \beta_5 T_{i,5} + \gamma X_i + \epsilon_i \qquad (2)$$

where $Y_i$ is a dummy capturing whether the respondent stays home and equal to 1 when the respondent answers 0 to a question on the amount of time spent outside home. The question specifies that by "outside one's home" it means "outside one's property". Being in the home garden, for instance, should not be considered being out of one's home. We estimate the same model twice, first on intended outcomes from the baseline survey and then on realised outcomes from the follow-up. $T_1$-$T_5$ are dummies equal to 1 if subject $i$ was randomly assigned to each of the 5 treatments ($T_{i,1}$ = You, $T_{i,2}$ = Family, $T_{i,3}$ = Others, $T_{i,4}$ = Country, $T_5$ = No framing/ Generic), and, is the error term. The first four dummies are further split in two when we separate the gain from the loss domain ($T_{i,1L}$ = You (Loss), $T_{i,1G}$ = You (Gain), $T_{i,2L}$ = Family (Loss), . . ., $T_5$ = No frame/ Generic). $X_i$ is a vector of covariates used at the randomisation stage to ensure balance between the groups. The variables included are gender, age, education, region, and household disposable income per capita. The control group is composed of people who do not receive any reminder. Hence, $\beta_0$ captures the proportion of people in the control group who intend to stay home (in the estimation on intentions) or actually do (in the estimation on subsequent actions). When we split the analysis by the health status of respondents, we divide the sample in two groups and estimate the model above separately on each of them.

## Results

### Impacts on intentions and actions

Our first finding is that the reminder significantly increases respondents' intentions to stay home when it is framed with respect to personal consequences and consequences for one's family. We test the null hypothesis of no difference between the control and each treatment group using standard significance testing. To this end, we estimate the empirical model outlined in the section "Estimation Strategy". For ease of exposition, we report treatment effects as percentage changes relative to the control group, together with the corresponding p-values and F-statistics. Fig 2 summarises the results, while Table 2 shows the full set of estimates. For simplicity, it reports them as percent increases relative to the share of people who stay home in the control group, which is just below 15% (i.e., an effect of 20 percent amounts to a 3-percentage point increase in the share of people who stay home, or 3 people out of 100). With both the "you" and the "family" framing, the treatment effect amounts to an increase in the share of people who intend to stay home of about 46% (p = 0.007 and p = 0.008, respectively) compared to the control group. On the other hand, the reminders have insignificant effects on intentions to stay home when they are framed with respect to the consequences for other people in general (12% with p = 0.459) and for the country as a whole (26.6% with p = 0.11). Similarly, the reminder with no framing—akin to the slogans commonly seen on social media (e.g., # STAY-HOME) and promoted by governments around the world (e.g., the SMS sent by the UK government and the Danish Police (Source: The Local DK (March 24, 2020) [link]; UK Government website (March 24, 2020) [link].)—has a statistically insignificant effect (26%, p = 0.19). These findings are in line with the hypothesis that emotional proximity to the people affected by the respondent's actions plays a strong role in determining the success of a message. They are also consistent with the findings of an interesting recent literature that uses experimental methods to study pro-social behaviour as a driver of health behaviour, including social distancing ([3, 26, 34]). We complement those findings by showing that individuals who comply with social distancing to protect others may do so primarily to protect their family.

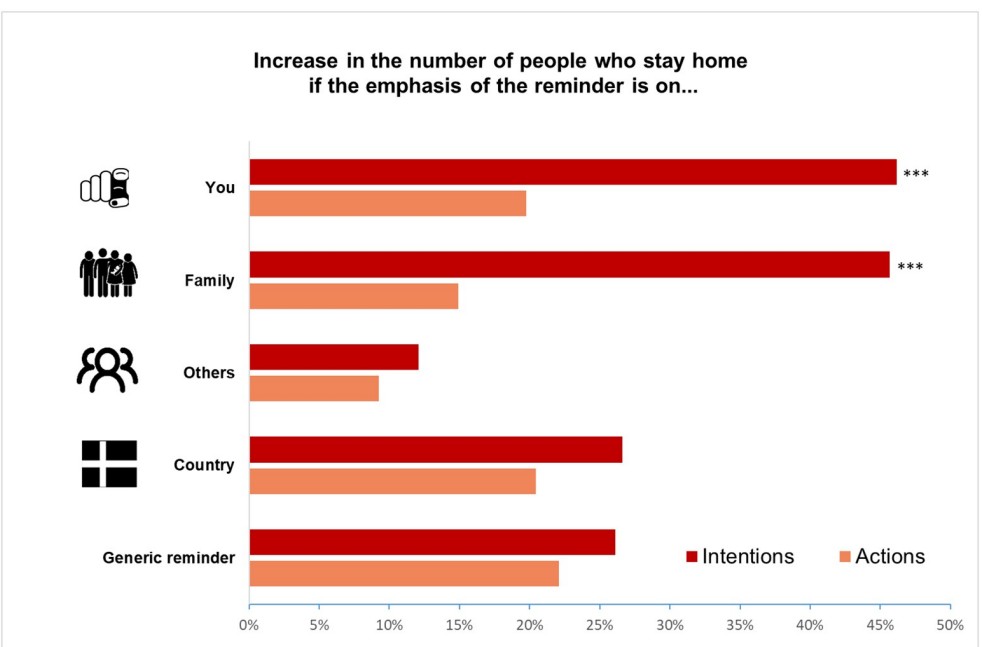

**Fig 2. The effects of different reminders on intentions and actions.** *Notes*: The effects are percentage changes relative to the share of people who intend to stay home or stayed home in the control group (i.e., the regression coefficients in Table 2 are divided by the share of people who intend to stay home (intentions) and stayed home (actions) in the control group). Intentions refer to the day after the first interview, actions refer to the day before the follow-up interview. The reminder increases respondents' intentions to stay home by 46% when it is framed with respect to personal consequences (p = 0.007) and consequences for one's family (p = 0.008). It has a lower insignificant effect on intentions when it refers to consequences for other people in general (p = 0.459), for the country as a whole (p = 0.110), and when it has no specific framing (p = 0.190). Changes in intentions do not translate into sizeable changes in actions. The reminders with a focus on personal consequences and consequences for one's family only increase the share of people who stay home by 19.7% (p = 0.127) and 14.9% (p = 0.251), respectively. As for intentions, the reminders have no significant impact on actions when they focus on "others" (p = 0.467), "country" (p = 0.113), or have no framing (p = 0.15). Respondents who referred to a weekday in the first interview and to a weekend day in the follow-up interview (and vice-versa) are dropped from the sample to avoid inconsistencies. The resulting sample size is N = 5,310. Stars reported at the top of the bars express the level of significance of the coefficient (*** p<0.01, ** p<0.05, * p<0.1).

Our second result is that when we further break down the most effective treatments ("you" and "family") and look at the sub-treatments in the gain and loss domain separately, we find very similar impacts independently of the domain. The estimated treatment effect on intentions ranges from 42% (p = 0.044) for the "family" treatment in the gain domain to 49% (p = 0.017) for the "family" treatment in the loss domain. The other treatments ("others" and "country") have no statistically significant impacts on intentions neither in the gain nor in the loss domain. The full set of results is available in Table 2. In general, we find that framing the reminders in terms of gains or losses has little impact on the results. This may be due to the fact that the differences between our alternative framings are too subtle to trigger the behavioural patterns predicted by prospect theory. Some respondents are reminded that going out may lead them to become infected (a loss). Some other respondents are reminded that staying home would prevent them from getting infected (a gain, *but only relative to a hypothetical situation in which going out leads to becoming infected*). Testing these differences was valuable to inform policymaking, but further work will be necessary to dig deeper into the framing effects predicted by prospect theory in this context.

**Table 2. Effect of treatments and sub-treatments on staying home (intention vs. action).**

| VARIABLES | (1) | (2) | (3) | (4) |
|---|---|---|---|---|
| | Intention | Action | Intention | Action |
| *You* | 0.044*** | 0.0288 | | |
| | (0.016) | (0.0188) | | |
| *Family* | 0.044*** | 0.0217 | | |
| | (0.016) | (0.0189) | | |
| *Others* | 0.012 | 0.0135 | | |
| | (0.016) | (0.0185) | | |
| *Country* | 0.025 | 0.0298 | | |
| | (0.016) | (0.0188) | | |
| *You loss* | | | 0.0449** | 0.0273 |
| | | | (0.0196) | (0.0223) |
| *You gain* | | | 0.0432** | 0.0302 |
| | | | (0.0193) | (0.0221) |
| *Family loss* | | | 0.0471** | 0.0251 |
| | | | (0.0198) | (0.0223) |
| *Family gain* | | | 0.0398** | 0.0183 |
| | | | (0.0197) | (0.0223) |
| *Others loss* | | | 0.00421 | -0.000453 |
| | | | (0.0181) | (0.0214) |
| *Others gain* | | | 0.0181 | 0.0260 |
| | | | (0.0181) | (0.0216) |
| *Country loss* | | | 0.0270 | 0.0171 |
| | | | (0.0185) | (0.0215) |
| *Country gain* | | | 0.0237 | 0.0436* |
| | | | (0.0189) | (0.0227) |
| *Generic* | 0.025 | 0.0322 | 0.0249 | 0.0322 |
| | (0.019) | (0.0223) | (0.0190) | (0.0224) |
| Controls | Yes | Yes | Yes | Yes |
| Observations | 5,310 | 5,310 | 5,310 | 5,310 |

*Notes*: The table shows the effect of receiving each reminder on the probability of staying home (relative to receiving no reminder). Control mean (intentions) = 0.0953; Control mean (actions) = 0.1457. Intentions refer to the day after the first interview, actions refer to the day before the follow-up interview. Respondents who referred to a weekday in the first interview and to a weekend day in the follow-up interview (and vice-versa) are dropped from the sample to avoid inconsistencies. The resulting sample size is N = 5,310. Controls include the following balancing covariates (used at the randomisation stage): gender, age, region, education, and household disposable income per capita. Robust standard errors in parentheses.

Confidence:

*** $p<0.01$,

** $p<0.05$,

* $p<0.10$.

Our third result is that changes in intentions do not translate into **equivalent** changes in actions. The two most effective treatments identified above ("you" and "family")—with effects on intentions of over 45%—only result into a 19.7% (p = 0.127) and a 14.9% (p = 0.251) increase in the share of subjects who actually stay home relative to the control group. The other treatments, which had lower insignificant impacts on intentions have even lower impacts on actions. Since detecting the potential significance of lower effects on actions is statistically more difficult, we run some robustness checks by aggregating affine treatments (as detailed in

Appendix D in S1 Appendix). Even then, we are unable to detect significant impacts on behaviour. When we test the joint hypothesis that all the reminders have an effect on actions equal to zero, we cannot reject it (F(5, 5299) = 0.75, p = 0.589). We also cannot reject the joint hypothesis that all the sub-treatments (considering the loss and the gain domain separately) have no impact on actions (F(9, 5295) = 0.75, p = 0.659).

Coupled with the first result, this evidence confirms that intention-to-action gaps can limit the effectiveness of messaging campaigns of this kind ([31, 62]), a possibility that has received limited attention in the existing literature on the impact of reminders during the COVID-19 pandemic ([23–26, 63]). Such gaps may be due to systematic behavioural biases (e.g., time inconsistency, planning fallacy), or idiosyncratic shocks forcing people to deviate from their intentions. While explaining such mechanisms is beyond the scope of this paper, documenting the divergence between intentions and actions is crucial for our understanding of how effective reminders are. In Appendix D in S1 Appendix, we run a battery of robustness checks, which lend further support to our conclusions.

One concern with the results is that the size of our sample poses challenges for statistical power and makes it difficult to detect a statistically significant effect on actions. When assessing this possibility, it is important to remark that the effect sizes we estimate on actions are small. Small effects on social distancing behaviour are consistent with recent work conducted during the pandemic ([64]). Since compliance in the control group is below 15%, an increase of approximately 20% in the probability of staying home corresponds to a change of less than 3 percentage points (or 3 people every 100). Detecting such a small impact would pose a statistical challenge even with a larger sample, which was beyond our possibilities. On the other hand, it seems reasonable to argue that an effect of such magnitude would be considered of limited societal relevance by many policy-makers, even if estimated more precisely. Finally, our conclusions are robust to the possibility that respondents may over-report staying home due to experimenter demand effects ([37]). Indeed, despite such potential over-reporting, we are unable to detect sizeable impacts on reported behaviour.

## Healthier people are harder to convince

The effects of the reminders may vary across social groups and many would argue that, in order to increase their effectiveness, they should be targeted at those who are least likely to comply with the recommendation. Previous research, for instance, finds that people who face the lowest risks from being infected are the most likely to diverge from social distancing measures during the COVID-19 pandemic ([23]). Are such groups responsive to the reminders? To answer this question, we split subjects according to their health status and repeat the analysis on separate samples. The lowest two (out of five) values of a variable indicating health status are considered bad health conditions for the purpose of this analysis (this is the categorisation that appears to be most sensible, since only considering in bad health those with the lowest value would leave us with a very small sample size). Fig 3 shows the results for the two most powerful treatments. Table 3 reports the full set of estimates.

The results indicate that respondents who are in worse health conditions and face the greatest risks from an infection are the most affected by the treatments. In particular, being reminded of the risks of going out for their families more than doubles their intended probability to stay home (p = 0.036) and increases their probability of actually staying home by over 80% (p = 0.034). These effects are even stronger if we focus on people with poor health who are relatively old (50 and above) (Not shown for conciseness).

The strong effect of the "family" reminder among people in bad health may be due to the fact that it reminds them of the burden and suffering that a worsening of their health

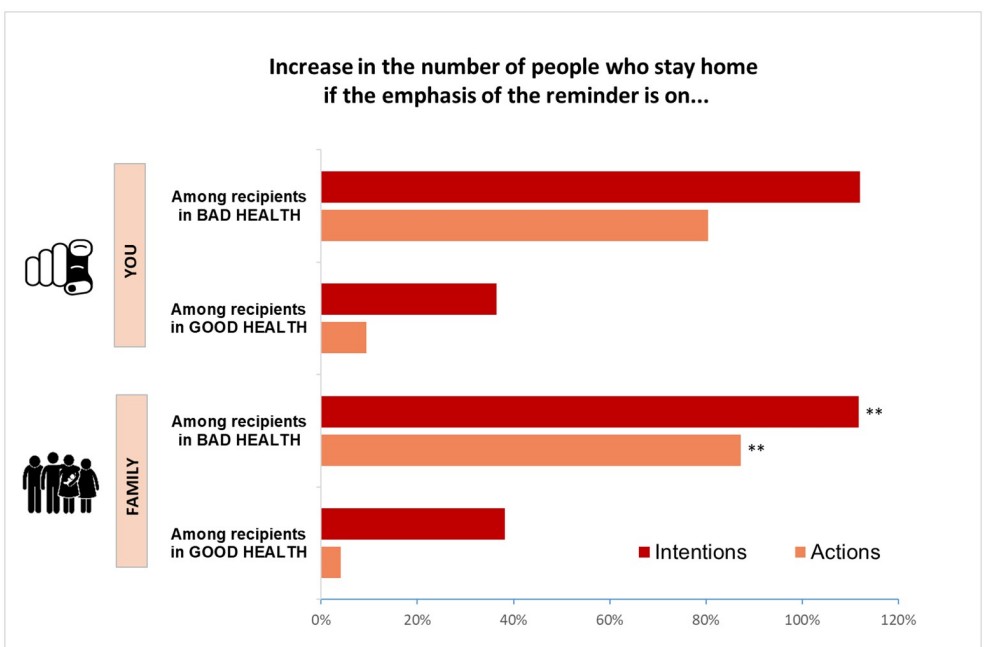

**Fig 3. Treatment effects of the most effective reminders by the health status of the recipient.** *Notes*: The effects are percentage changes relative to the probability of staying home in the control group, which receives no reminder (i.e., the regression coefficients in Table 3 are divided by the probability of staying home in the control group). Among respondents who are in bad health conditions (N = 603), the share of those who declare they will stay home more than doubles after receiving a reminder that emphasises risks for family (p = 0.036), and the share of those who actually stay home increases by 80% (p = 0.034). Similar impacts of the "you" treatment are not statistically significant. Among respondents who are in good health (N = 4,704), the impacts of the reminder are much smaller and not statistically significant. It is important to note that while impacts of over 30% may appear sizeable, they a relative to a low compliance level in the control group (where less than 15% of respondents stay home). Respondents classify their health status on a 5-point scale. The lowest two values are considered bad health conditions for the purpose of this analysis (health information is missing for 3 observations used in the main regressions). Stars reported at the top of the bars express the level of significance of the coefficient (*** p<0.01, ** p<0.05, * p<0.1).

conditions would impose on their loved ones. Another plausible explanation is that subjects in poor health live with other people in similar health conditions (e.g., older couples).

On the other hand, people with better health, who face the lowest risks from an infection (and are the ones who go out of their homes the most) are not affected by the reminders. These results show that reminders may help to protect groups at risk by increasing their likelihood of staying home, while they do not increase compliance among those who face limited personal health risks but may spread the disease.

## Robustness checks

In this section, we document the results of a battery of robustness checks to test the sensitivity of our results with respect to some key choices made when defining the sample of interest. The results are reported in Tables 8, 9 and Appendix D in S1 Appendix.

First, the sample for the analysis was limited to respondents who referred to a weekday or a weekend day both in the baseline and in the follow-up survey (Table 8 in S1 Appendix, Col.1–2). This is a reasonable approach, as not doing so would make the two answers incomparable. Nonetheless, we check the robustness of our results to reinstating all the observations we have. Upon doing that, we find that the magnitude of the effects of the most effective treatments ("you" and "family") decreases slightly, but treated subjects still declare an intention to stay

**Table 3. Heterogeneous effects by health status.**

| | Good Health | | Bad Health | |
|---|---|---|---|---|
| **VARIABLES** | **(1)** | **(2)** | **(3)** | **(4)** |
| | **Intention** | **Action** | **Intention** | **Action** |
| *You loss* | 0.033* | 0.0131 | 0.157* | 0.161* |
| | (0.020) | (0.0227) | (0.0804) | (0.0877) |
| *You gain* | 0.044** | 0.0270 | 0.0410 | 0.0557 |
| | (0.020) | (0.0229) | (0.0666) | (0.0762) |
| *Family loss* | 0.034* | 0.00574 | 0.156** | 0.174** |
| | (0.020) | (0.0226) | (0.0742) | (0.0821) |
| *Family gain* | 0.039* | 0.0158 | 0.0490 | 0.0386 |
| | (0.020) | (0.0232) | (0.0689) | (0.0775) |
| *Others loss* | -0.005 | -0.0154 | 0.0856 | 0.124 |
| | (0.018) | (0.0217) | (0.0704) | (0.0810) |
| *Others gain* | 0.003 | 0.00900 | 0.129* | 0.149* |
| | (0.018) | (0.0220) | (0.0672) | (0.0763) |
| *Country loss* | 0.012 | 0.00916 | 0.113* | 0.0541 |
| | (0.019) | (0.0223) | (0.0668) | (0.0714) |
| *Country gain* | 0.016 | 0.0302 | 0.0785 | 0.144* |
| | (0.019) | (0.0234) | (0.0693) | (0.0800) |
| *Generic* | 0.017 | 0.0230 | 0.0747 | 0.0922 |
| | (0.019) | (0.0231) | (0.0683) | (0.0755) |
| Controls | Yes | Yes | Yes | Yes |
| Observations | 4,704 | 4,704 | 603 | 603 |

*Notes*: The table shows the effect of receiving each reminder on the probability of staying home (relative to receiving no reminder). Control mean—Good Health (intentions) = 0.089; Control mean—Good Health (actions) = 0.138; Control mean—Bad Health (intentions) = 0.14; Control mean—Bad Health (actions) = 0.2. Respondents classify their health status on a 5-point scale. The lowest two values are considered bad health conditions for the purpose of this analysis (focusing exclusively on those with the lowest value would leave us with little statistical power). Health information missing for 3 respondents. Controls include the following balancing covariates (used at the randomisation stage): gender, age, region, education, and household disposable income per capita. Robust standard errors in parentheses.

Confidence:

*** $p < 0.01$,

** $p < 0.05$,

* $p < 0.10$.

home 31% more ("Family" treatment in the loss domain) than subjects in the control group (whose likelihood to stay home is 10.6%). Column 2 confirms that the effect sizes get smaller when we turn to actions. To further test whether potential mismatches in days of the week between the first and the second interview influence our findings, we control for day of the week in our specifications as an additional robustness test and the results do not change (Table 9 in S1 Appendix). Random assignment to different treatment groups further ensures that problems of this kind do not play a role, since they are likely to affect all respondents irrespective of the reminder they receive. The same reasoning applies to other potentially confounding factors, such as weather conditions (which may impact whether people go out or stay home). The balance tests in Table 4 in S1 Appendix confirms that the sample is balanced across treatments (including in the geographical distribution of respondents, which implies balance in weather conditions).

Second, we test how our results change when we drop respondents who declared an intention to spend (or having spent) 24 hours outside their home (Col.3–4). Such responses are genuinely difficult to interpret. Those people may be away from their home for several days and may well have isolated themselves where they are (e.g., at a vacation house, which would not seem unlikely at the time of the experiment given the good weather), despite having reported being away from home for the entire day. Whether we drop such observations or code them as if the respondents stayed home, our results do not change.

Third, certain respondents answer the follow-up questionnaire with a significant delay after having responded to the baseline survey (Col.5–6). This poses potential concerns regarding the comparability of their answer in the first questionnaire with their answers in the second one. When we confine the analysis to respondents who complete the follow-up questionnaire within one week from the baseline, both the magnitude and the statistical significance of the estimated treatment effects increase slightly for the most effective treatments ("you" and "family"), but our conclusions do not change.

Fourth, since the COVID-19 crisis evolved very quickly and the situation changed between the first and the second week in our study period (as the Danish government gave the first signs of wishing to relax the restrictions it had imposed), we also test how the results change when we separate subjects who responded within the first week from the rest (Col. 7–8). When we drop the latter (who responded at a time when the situation was starting to become less tense in Denmark), our conclusions do not change. In fact, the effect on intentions of the treatment framed with respect to the dangers for one's family becomes even stronger and remains statistically significant despite the lower sample size. We still detect smaller effects on actions that are not statistically significant.

Finally, since the treatment effects on actions appear to be generally smaller than those on intentions, detecting their statistical significance is naturally more difficult. In order to increase statistical power, we pool the "you" and the "family" treatment and re-run the analysis on both intentions and actions. Such a strategy is inspired by the conceptual affinity of those two treatments (both pertaining to the personal sphere) and is corroborated by the fact that they have very similar effects on both intentions and actions throughout the analysis. For simplicity, we also pool the other two framed treatments ("others" and "country"), which are also conceptually affine (whether we do that or not, however, does not change the results). Upon running such a test (Col.9–10), we are unable to detect significant effects of the aggregate treatments on actions, despite the increase in statistical power, and our conclusions do not change.

## Discussion

This paper sheds new light on the effectiveness of messaging campaigns in promoting social distancing during the COVID-19 pandemic. It reveals that while reminders may be effective in changing people's intentions, those intentions are not matched by sizeable changes in subsequent actions. The conclusion is robust to the possibility that respondents may over-report compliance. If anything, that should artificially inflate our estimate impacts. Despite that, the effects we detect are small and statistically insignificant.

Our evidence indicates that intention-to-action gaps may be an important obstacle in the promotion of social distancing during a pandemic and that messaging campaigns are unlikely to be effective unless they tackle such gaps. This could be achieved, for instance, by increasing the frequency of reminders to reduce the burden of time inconsistency, though the benefits of such a strategy should be weighed against the risk of habituation. We also show that reminders are most effective in inducing behavioural change among people in relatively poor health, while subjects who are in good health are not affected. This is consistent with the idea that

reminders are meant to leverage people's prior convictions, rather than changing people's minds. This suggests that in order to induce behavioural change, reminders should be targeted at specific audiences.

Our findings bear important lessons for the international community. Messaging campaigns like the one we tested have been used extensively across the world and will continue to play an important role for the foreseeable future. Understanding what types of messages are most effective and being alert to the existence of important gaps between people's intentions and their actions will help to inform more effective messaging campaigns.

## Supporting information

**S1 Appendix.**
(PDF)

## Acknowledgments

We are grateful to Marco Piovesan, Claus Thustrup Kreiner, Rudi G.J. Westendorp, Laust Hvast Mortensen, Alexander Sebald, Davide Dragone, Pol Campos-Mercade, Christina Gravert, Mauro Caselli, Stefano Caria, Magnus Johansson, and all participants in the University of Copenhagen COVID-19 Seminar for their helpful comments. We are indebted to Bo Lønberg Bilde and his colleagues from DST Survey (Statistics Denmark) for their invaluable assistance and great efforts. We thank the Department of Economics, the Center for Economic Behaviour and Inequality (CEBI), the Department of Public Health ("CHALLENGE" project NNF17OC0027812), and the Centre for Healthy Ageing (CEHA) at the University of Copenhagen for their financial support. The activities of CEBI are financed by the Danish National Research Foundation, Grant DNRF134. We are thankful to Arash Bal, Gan Khoon Lay, Nikita Kozin, and Priyanka from the Noun Project for some of the graphics used in our charts. All errors are our own. Declarations of interest: none.

## Author Contributions

**Conceptualization:** Paolo Falco, Sarah Zaccagni.

**Data curation:** Paolo Falco, Sarah Zaccagni.

**Formal analysis:** Paolo Falco, Sarah Zaccagni.

**Funding acquisition:** Paolo Falco, Sarah Zaccagni.

**Investigation:** Paolo Falco, Sarah Zaccagni.

**Methodology:** Paolo Falco, Sarah Zaccagni.

**Project administration:** Paolo Falco, Sarah Zaccagni.

**Resources:** Paolo Falco, Sarah Zaccagni.

**Software:** Paolo Falco, Sarah Zaccagni.

**Supervision:** Paolo Falco, Sarah Zaccagni.

**Validation:** Paolo Falco, Sarah Zaccagni.

**Visualization:** Paolo Falco, Sarah Zaccagni.

**Writing – original draft:** Paolo Falco, Sarah Zaccagni.

**Writing – review & editing:** Paolo Falco, Sarah Zaccagni.

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
