## [Decision Letter · Decision Letter 0]

21 Sep 2021

PONE-D-21-25931Promoting social distancing in a pandemic: Beyond the good intentionsPLOS ONE

Dear Dr. Falco,

Thank you for submitting your manuscript to PLOS ONE. After careful consideration, we feel that it has merit but does not fully meet PLOS ONE’s publication criteria as it currently stands. Therefore, we invite you to submit a revised version of the manuscript that addresses the points raised during the review process.

We look forward to receiving your revised manuscript.

Kind regards,

Nikolaos Askitas

Academic Editor

PLOS ONE

Additional Editor Comments:

I would be happy to receive and evaluate a revision which responds to the comments by the reviewers.

Also here is a point which troubles me and I would like a response on. You say in the abstract: "This is despite the possibility that respondents may tend to over-report compliance."

My point is that you feel your results are shielded from over-reporting compliance because of obvious reasons. Let me propose though that the "planning fallacy" could be trouble here (https://en.wikipedia.org/wiki/Planning_fallacy).

Also you compare reports about things people say they will do "tomorrow" with things they say they did "yesterday" at a later point in time. From Fig. E.1 I suppose "tomorrow" and "yesterday" are different weekdays which opens you to day-of-week effects you need to account for. What happens if you control for day of week in your regression?

Also how far away from home people will go and how long that will last is affected by weather conditions which might differ across space and time. A comment on why this does not play a role here would appear to be necessary. If it could you might need to add weather in your regression (precipication, temperature, wind speeds should be easy to get).

Finally something minor. Please add the year in Fig. E.1. Years from now it will make the life of readers easier plus a Fig. should contain all the info necessary to fully understand it.

Journal Requirements:

Reviewers' comments:

Reviewer's Responses to Questions

**Comments to the Author**

1. Is the manuscript technically sound, and do the data support the conclusions?

Reviewer #1: Yes

Reviewer #2: Yes

Reviewer #3: Yes

2. Has the statistical analysis been performed appropriately and rigorously? 

Reviewer #1: Yes

Reviewer #2: Yes

Reviewer #3: Yes

3. Have the authors made all data underlying the findings in their manuscript fully available?

Reviewer #1: Yes

Reviewer #2: Yes

Reviewer #3: Yes

4. Is the manuscript presented in an intelligible fashion and written in standard English?

Reviewer #1: Yes

Reviewer #2: Yes

Reviewer #3: Yes

5. Review Comments to the Author

Reviewer #1: While the contribution is not revolutionary, nor the proposed solutions (from the experimental treatments) particularly strong, it is well-described and motivated and contributes to our understanding of how to improve COVID messaging around the world. Therefore I suggest acceptance with a few very minor revisions (some other comments below as well):

- maybe there is a difference b/t American and Italian English, but I think grammatically the subtitle would be more correct as "beyond good intentions"

- I would change the first line of the abstract to "Do reminders to promote social distancing achieve desired effects in behavior, not just intentions?" to highlight your contribution better

- nice experimental set-up with impressive sample size and compliance

- it wasn't totally clear from the main text that your validation with the apple data was aggregate and not individual-level; written right now to suggest it is individual-level -- please clarify this

- table 1: in my mind, the column names for frame and domain should be switched; e.g. "family" is a domain

- I found the robustness checks generally convincing

Reviewer #2: The manuscript details the results of an experiment of whether reminding people to comply with public health guidelines impacts their intentions and actual behaviors in the context of COVID-19. I think the results are interesting, novel, and timely.

My only real concern is that there is almost no theory to explain these results. I think the authors need to work to provide a theoretical foundation for their findings. Certainly, there is a significant amount of theory to work with from micro-economics and psychology on the efficacy of nudging experiments, and/or the utility of how narratives are framed.

Reviewer #3: This manuscript uses a messaging intervention to examine how to increase compliance with stay at home or social distancing behavior in Demark. The authors find that the message influences behavioral intention in the expected direction, but does not influence longer term behaviors. Overall, it’s a good paper – the study is competently designed, and the results are described accurately, so I’m happy to recommend relatively minor revisions.

One thing that might be nice to see is hypotheses delineated in the text more clearly. It’s obvious the authors have them (it’s an experiment and its pre-registered!), but it’s not really clear in text what they are before we get to the research design.

I found the discussion of prospect theory a bit lacking. The discussion of the literature on COVID-19 and public health measures is good, but prospect theory gets only kind of a passing citation to a 2013 review article. I’d like to see more here, especially since PT is so relevant to the design of the study (though, perhaps not to the analyses).

This brings me to a point that the authors can’t avoid, but one that I think merits discussion. The authors generally find no differences (either in intention or action) based on the loss vs. gain framing. Obviously, the authors are aware that PT predicts differences in risk seeking behavior on losses vs. gains – risk seeking in domain of losses, and risk aversion in the domain of gains. I would guess this translates to – more likely to stay home in gain frame, less likely in the loss frame.

Of course, I can see some explanations of why this might be the case. It doesn’t totally fit the PT framework – the risk individuals take by leaving home doesn’t seem to necessarily have a payoff with respect to public health outcomes. This is a divergence from the classic PT public health work, where the “risky’ option has direct positive public health consequences if the risk pays off. So, I see this less as a limitation to the work and more as a limitation to PT generally – if there are no clear and relevant benefits to the risk, loss/gain framing seems to have no impact on behaviors. I think this is definitely worth discussing, as it expands this manuscript from speaking to COVID-19/public health literature to speaking generally to PT and decision-making literature.

Another point worth making is that the differences between intentions and actions themselves are possibly not statistically different (though eyeballing it looks like it might be close!). This doesn’t change the quality of the authors’ work, but does probably change their interpretation a bit. I’m hesitant to say there are no behavioral effects here – all are in the expected direction, save 1 which is functionally zero, and all may have reasonable effect sizes (excuse the gross reference to my own work, but a paper from Roberts and Utych shows generally small effects on social distancing behavioral differences in the US -https://www.researchgate.net/publication/348607480_Polarized_Social_Distancing_Residents_of_Republican-Majority_Counties_Spend_More_Time_Away_from_Home_During_the_COVID-19_Crisis). The authors have quite a few observations, but they also have quite a few groups. They’re still at roughly 590 per group, which should be fine for detecting effects, but I’d be more comfortable saying the effect sizes get smaller, rather than disappear, given that these probably are small effects that would be distinguished in a very large sample (as is available with some behavioral data out there).

Overall, I hope I have conveyed that I have a positive assessment here, and that the authors are doing really good and well-designed work. I think this should be published, pending these small revisions.

Reviewer: Steve Utych

6. PLOS authors have the option to publish the peer review history of their article (what does this mean?). If published, this will include your full peer review and any attached files.

Reviewer #1: No

Reviewer #2: No

Reviewer #3: **Yes: **Stephen M. Utych

---

## [Author Response · Author response to Decision Letter 0]

5 Nov 2021

Dear Dr. Askitas,

As requested, we have provided point-by-point responses to the comments received in an attached document (Falco & Zaccagni_Response to Reviewers.pdf).

Best regards,

Paolo Falco and Sarah Zaccagni

---

## [Editor Report · Decision Letter 1]

10 Nov 2021

Promoting social distancing in a pandemic: Beyond good intentions

PONE-D-21-25931R1

Dear Dr. Falco,

We’re pleased to inform you that your manuscript has been judged scientifically suitable for publication and will be formally accepted for publication once it meets all outstanding technical requirements.

Kind regards,

Nikolaos Askitas

Academic Editor

PLOS ONE

Additional Editor Comments (optional):

I am very satisfied with your responses to reviewers' and editor's comments. Thank you for being thorough. This is a very well done paper indeed.
---

## [Editor Report · Acceptance letter]

19 Nov 2021

PONE-D-21-25931R1 

Promoting social distancing in a pandemic: Beyond good intentions 

Dear Dr. Falco:

I'm pleased to inform you that your manuscript has been deemed suitable for publication in PLOS ONE. Congratulations! Your manuscript is now with our production department. 

Kind regards, 

on behalf of

Dr. Nikolaos Askitas 

Academic Editor

PLOS ONE